# COVID-19 Vaccine Mandates: Attitudes and Effects on Holdouts in a Large Australian University Population

**DOI:** 10.3390/ijerph191610130

**Published:** 2022-08-16

**Authors:** Katie Attwell, Leah Roberts, Julie Ji

**Affiliations:** 1School of Social Science, University of Western Australia, Crawley, WA 6009, Australia; 2Wesfarmers Centre of Vaccines and Infectious Diseases, Telethon Kids Institute, Subiaco, WA 6008, Australia; 3Immunisation Alliance of Western Australia, Cockburn Integrated Health and Community Facility, Suite 14, 11 Wentworth Parade, Success, WA 6164, Australia; 4School of Psychological Science, University of Western Australia, Crawley, WA 6009, Australia

**Keywords:** mandatory vaccination, COVID-19, university, mixed method, pandemic, qualitative, quantitative

## Abstract

Many governments and institutions mandated COVID-19 vaccines. In late 2021, we sought to ascertain the perspectives of staff and students from The University of Western Australia about the State or the University mandating COVID-19 vaccines. The survey captured vaccination status and intentions along with attitudes towards mandates and potential types of exemptions with 2878 valid responses which were quantitatively analysed and 2727 which were qualitatively analysed. The study found generally high levels of vaccination or intent, and strong support for mandates, underpinned by beliefs that vaccination is a moral duty and that mandates make campus feel safer. These sentiments were not more prevalent amongst individuals with comorbidities; often healthy individuals supported mandates to reduce their risk of transmitting disease to vulnerable family members. Individuals with comorbidities were, however, more supportive of excluding the unvaccinated from campus. Most opponents were unvaccinated, and many indicated that mandate policies would backfire, making them less likely to vaccinate. Despite the strong overall support, 41% of respondents did not want to see non-compliant staff or students lose their positions, and only 35% actively sought this. Institutions or governments introducing mandates should emphasise community concerns about catching COVID-19 and becoming sick or transmitting the disease to vulnerable loved ones.

## 1. Introduction

With the advent of the COVID-19 pandemic, governments and institutions have sought to attain high vaccine uptake. Mandates have been used globally in a variety of settings for decades to increase uptake of vaccines by imposing consequences for non-vaccination [1]. Many governments and institutions globally have mandated COVID-19 vaccines for populations or workforces [2]. We define vaccine mandates as interventions imposing consequences for non-vaccination. Relevant consequences imposed by governments or organisations can include criminalised non-vaccination [3], withdrawal of financial benefits [4], exclusion from public goods and public spaces [5] and restrictions on travel, as seen during the COVID-19 pandemic [6]. Vaccine mandates vary widely in their ‘salience’ to make people vaccinate, with structured exemption processes available in some regimes enabling people to not vaccinate for non-medical reasons, including personal belief or religion [5]. Such exemptions are common in the United States and some European countries, but are not widely used elsewhere [7].

Australian institutions are familiar with vaccine mandates, which state and federal governments employ to promote childhood vaccine uptake [4]. Australian mandates do not permit philosophical or religious exemptions. Mandates for childhood vaccines and COVID-19 vaccines are widely supported [8,9] but there have been no quantitative studies of attitudes towards mandates since a number of state governments introduced mandates for occupations or social life (attendance at sporting games, hospitality venues etc.). Recent qualitative work suggests that public understandings of COVID-19 mandates are nuanced and sometimes confused, reflecting the diversity of the policies themselves. Medical exemptions are well-understood and supported, but the public gives short shrift to personal belief or religious exemptions [2].

Western Australia (WA) experienced a unique position during the COVID-19 pandemic. Due to the early closure of the international border and the erection of hard state borders, WA experienced little to no community transmission of COVID. In the first year of the pandemic, just over a thousand cases of COVID-19 were reported [10]. The majority of cases occurred within hotel quarantine where returned travellers were placed for two weeks. The pandemic did not break out in the state until early 2022, by which time 90% of the population aged over 12 had received two vaccine doses and the state was about to reopen on 3 February [11]. The reopening was delayed due to Omicron waves in other states and the desire to allow more of the population to access third (or, for children aged 5–11, first) doses. There was considerable public, academic and policymaker interest in how this third dose would be received by the population and whether it would also be mandated. The virus had breached the state borders by the eventual reopening on 3rd March, and low-scale community transmission was already occurring.

Despite a lack of community transmission throughout 2020 and 2021, WA’s State government introduced sweeping vaccine mandates late in 2021 to cover 75% of the workforce [12]. However, they did not include universities, meaning these large institutions had to develop their own policies regarding admitting unvaccinated staff and students onto campus, or allowing them to retain their employment or enrolment.

Universities elsewhere introduced their own COVID-19 vaccine mandates for students and staff. In the United States, a number of universities required vaccination to attend campus [13]. As of January 2022, Harvard University required all eligible community members to be vaccinated and to have had the booster if eligible [14]. Some Canadian [15] as well as many European universities also required this [16]. In Australia, University of Melbourne [17] and a number of universities in New South Wales required vaccination [18].

In late 2021, ahead of WA’s reopening and while there was still no community transmission, The University of Western Australia (UWA) sought to understand the viability and acceptability of introducing a COVID-19 vaccine mandate covering all students and staff. Although universities had been excluded from recent State Government mandates, there remained the possibility that the University may yet be subject to an additional mandate under a government Public Health Order. With no previous formal research studies undertaken to assess the public acceptability of mandates in a large university, we designed and conducted a mixed methods survey of students and staff to pursue novel research questions pertinent to vaccine mandates and vaccine acceptance. We sought to ascertain participants’ perspectives via quantitative and qualitative instruments; the same broad survey captured both cohorts’ attitudes towards COVID-19 vaccination and prospective State Government or UWA vaccine mandates, vaccination status and intentions to be vaccinated against COVID-19, or intentions to remain unvaccinated and why. Particular interest was taken in participants with comorbidities who are more vulnerable to catching COVID-19 and becoming seriously ill. We were similarly interested in whether people who had accepted two vaccine doses would be willing to have a third. The survey sought to understand attitudes towards exemptions that may be offered, how participants thought refusal to vaccinate should be treated, and whether additional measures could enhance vaccine confidence and uptake. We intended our findings to be useful to governments and institutions faced with making decisions about whether and how to mandate COVID-19 vaccinations.

## 2. Methods

### 2.1. Study Design, Ethics, and Consent Processes

The research team modified a survey on childhood vaccine mandate attitudes used in a study of school administrators in Michigan, USA (unpublished), also drawing from materials shared by the University of Adelaide, which surveyed its own staff and student cohorts in late 2021 in a non-academic study. Questions were adapted for the local context drawing upon findings from the Coronavax Project, a mixed methods study of the West Australian population’s attitudes and information needs regarding COVID-19 vaccinations. which the first author leads. The survey questions reflect independent standalone questions that directly elicited participants’ opinions at a specific time window during the pandemic in Western Australia. These standalone survey questions were not part of a questionnaire instrument designed to use multiple questions to assess an underlying construct that is stable across time. The University’s COVID-19 Vaccine Mandate Working Group consulted on the study design, inviting feedback from the National Tertiary Education Union and student representatives. Survey questions are included in the Appendix A.

During the study period, the University distributed the voluntary anonymous survey to 6490 staff and 21,543 students. Distribution channels included email, newsletters, and computer lock screens. Participation was incentivised through an AUD 50 voucher draw, with four vouchers available to staff, and four for students. UWA’s Human Research Ethics Committee provided approval (2021/ET001082). Participants viewed an information sheet at the commencement of the survey and consented to participate by continuing; they were free to withdraw by closing the computer browser.

### 2.2. Data Collection and Processing

The survey was administered online via the survey platform Qualtrics using the University of Western Australia Qualtrics account, therefore displayed the official UWA logo. The survey was disseminated internally via UWA student and staff emails by the UWA Office of the Vice-Chancellor. All questions required a response to be made, except when open-ended comments were elicited. There was no option to revise a response once submitted. The survey was open from 29 November to 8 December 2021, although publicity materials stated that the survey would close at the end of 3 December. As a number of respondents completed the survey after that date, all valid responses were included in the quantitative study. A total of 3433 responses were collected; 2930 were identified as complete; 503 incomplete responses were excluded; 52 responses were deemed likely duplications due to being recorded from the same IP address as well as having the same age, gender, employment type, and campus location. With an anonymous survey, it was not possible to verify whether such responses were duplicates or unique entries from different respondents, so they were excluded. The third author quantitatively analysed 2878 valid responses. Statistical analysis involved testing for group differences using the chi-square test of independence using the statistical software package Jamovi. The final sample represents an approximate response rate of 8.50% for students and 25% for staff.

In total, 2727 of the 2878 responses were qualitatively analysed, reflecting those collected at the official deadline of 5 December, with all responses imported into NVivo from the larger survey’s Excel spreadsheet output. The intention of the qualitative analysis was to ascertain reasons for participants opposing university or government mandates that they would state in their own words, without cues from a multiple choice survey. Accordingly, qualitative data were drawn from the entirety of participants’ answers to three open-ended questions within the survey: (1) If participants endorsed “strongly opposed”, “somewhat opposed”, or “neutral” regarding their stance on the imposition of a *UWA vaccine mandate*, they were asked to provide reasons for their stance; (2) If participants endorsed “strongly opposed”, “somewhat opposed”, or “neutral” regarding the imposition of a *government vaccine mandate*, they were asked to provide reasons; (3) A further comments section at the end of the survey invited *all respondents* to provide answers, which we included as an important opportunity to hear participants’ voices, especially since this was the only opportunity for mandate supporters to provide free text answers. 1566 of 2727 respondents provided non-blank responses here: 609 substantial comments were analysed and are reported in this paper; a further 292 simple statements such as ‘Do it!’ were not further analysed; the rest of the comments which did not contain any relevant content included redundant remarks such as “thank you” or “nothing further to add”. A thematic analysis of the 609 substantial comments was conducted in NVivo 20 by the second author [19] in regular consultation with the lead author. Survey responses were initially coded deductively based on reasons participants were ‘neutral’ or ‘opposed’ to mandates with categories developed from existing literature, particularly ethical and policy literature [20,21], and the coding tree was augmented iteratively through inductive coding. As we were interested in the prevalence of particular reasons, we coded the entire dataset rather than considering saturation. Further comments were coded inductively and, again, in their entirety. For all qualitative analysis, representative quotes were selected that best epitomised an argument or position coded, and these are reproduced below. All names are pseudonyms and privacy of personal data has been strictly maintained.

## 3. Results

### 3.1. Participant Characteristics

Demographics information for the full sample of 2878 respondents are summarised below and presented in Appendix A.

**Gender.** 1635 respondents identified as female (56.81%), compared to 1169 as male (40.62%), 33 as non-binary or third gender (1.15%); 41 preferred not to say (1.42%).

**Age.** Most respondents were aged under 35 years of age, with 10 aged under 18 (0.35%), 1158 aged between 18–24 (40.24%), 541 aged between 25–34 (18.80%), 445 aged between 35–44 (15.46%), 347 aged between 45–54 (12.06%), 260 aged between 55–64 (9.03%), 85 aged between 65–74 (2.95%), 8 aged between 75–84 (0.28%), and 4 aged 85 or older (0.14%); 20 respondents did not report their age (0.69%).

**Occupation.** Most respondents were students, with 1030 undergraduates (35.79%) and 807 postgraduates (28.04%); 565 respondents identified as professional staff (19.63%), and 392 as academic staff (13.62%). An additional 84 respondents identified as “Other” (2.92%), comprising adjunct, emeritus, or honorary staff, visitors, medical/health/government facility staff, alumni, and family members of students.

**Underlying health condition.** In total, 259 respondents (9%) reported an underlying health condition that made them more susceptible to COVID-19. A comprehensive list of self-reported health conditions is provided in Appendix A; these have not been clinically verified as (a) true or (b) constituting a recognised comorbidity for COVID-19.

#### 3.1.1. Vaccination Status and Categorisation

Respondents were classified based on their responses to the question concerning their vaccination status for COVID-19, as reported in Table 1 below. We categorised participants as “double vaccinated or willing to be” to measure compliance (or intended compliance) based on criteria for full vaccination at the time. We classified the 31 people willing to have a first dose as willing to be fully vaccinated (double vaccinated) as such individuals did not endorse the other options that indicated hesitancy (i.e., “Will not get vaxed”, “Undecided about vax”, “Vax status not disclosed”).

In total, 2662 respondents (92.49%) were classified as “Double vaccinated or willing to be”. In contrast, 216 (7.51%) were classified as “Not double vaccinated or willing to be”. We sought to understand whether the 2545 double vaccinated participants were willing to accept a booster; 2324 (91.32%) were willing, 199 (7.82%) were unsure, and 22 (0.86%) were unwilling. This means that 8.68% of our sample who were fully vaccinated at the time of the study were nevertheless reticent to accept a booster dose.

#### 3.1.2. Vaccination Status by Demographic Variables

Participants’ vaccination status (whether they are double vaccinated or willing to be versus not willing to be) as a function of their demographic variables are reported in Appendix A.

**Gender.** Respondents’ vaccination status did not differ as a function of gender (female vs. male), chi-square (2, *N* = 2837) = 0.078.

**Age.** Respondents’ vaccination status were not statistically different for the main age categories (18–24 vs. 25–34 vs. 35–44 vs. 45–54 vs. 55–64), chi-square (4, *N* = 2751) = 7.32, *p* = 0.120.

**Occupation.** Respondents’ vaccination status differed between staff and students, chi-square (3, *N* = 2794) = 28.3, *p* < 0.001, with double vaccinated or willing to be status being most prevalent in academic (98.50%) and professional staff (94.70%) as compared to postgraduate (91.80%) and undergraduate students (91.1%).

**Health status.** Vaccination status did not differ between those who reported an underlying health condition versus those who did not, chi-square (1, *N* = 2878) = 0.527.

### 3.2. University Vaccine Mandate Stance and Categorisation

#### 3.2.1. University Mandate

We asked our respondents “In the absence of a government mandate, would you support a mandate by the University that requires everyone attending campus to be vaccinated, if they are able to be”? Of the 2878 respondents, 1756 (61%) strongly supported and 529 (18.4%) somewhat supported a university mandate. In contrast, 331 (11.5%) of respondents strongly opposed and 107 (3.7%) somewhat opposed one. A further 155 (5.4%) of respondents were on the fence (neither supported nor opposed).

Table 2, below, simplifies this into three categories—Support, Oppose, Neutral. Almost 80% of respondents supported a UWA mandate, as compared to 15.22% who opposed it.

#### 3.2.2. University Vaccine Mandate Stance as a Function of Vaccination Status

We would expect opposition to vaccine mandates to align closely with individuals being unvaccinated or not intending to have further doses, as the policy would adversely affect them. However, we would also expect some vaccinated persons to oppose or be neutral about mandates, since being vaccinated does mean that one supports all policy mechanisms for achieving this status in oneself or others.

Whether respondents supported or opposed a university vaccine mandate did differ significantly depending on whether they were double vaccinated or willing to be, or not, chi-square (2, *N* = 2878) = 1165, *p* < 0.001. Of the 2662 respondents who were double vaccinated or willing to be (DVWTB), 2280 supported a university mandate (85.65%), 232 opposed one, (8.72%), and 150 were neutral (5.63%). In contrast, of the 216 respondents who were Not DVWTB, only 5 supported a university mandate (2.31%), 206 opposed one, (95.37%), and 5 were neutral (2.31%).

Many participants reiterated their support for UWA mandates in free text at the end of the survey; some offering further reasons for their perspectives:
*“It seems like the reasonable thing to do, to ensure the safety of all members of the UWA community”.*(John, postgraduate, vaccinated)
*“I think that mandates are necessary, especially as with the borders reopening”.*(Sarah, professional staff, vaccinated)
*“I strongly believe they should, you have a campus of 30–40,000 people to not mandate it because some students/staff have alternative views whilst accommodating is actually endangering thousands let alone the hundreds if not thousands of students/staff that would be within the high-risk categories and thus is irresponsible”.*(Tom, undergraduate, vaccinated)

Of the small number of participants who were not vaccinated or did not disclose their vaccination status but nevertheless supported UWA mandates, none provided reasons. We asked the 155 participants who were neutral about UWA mandates to provide reasons. The 67 who provided substantive reasons reported feeling conflicted generally about mandates and raised concerns around freedom and implementation.
*“I support vaccination… but feel uncomfortable with the idea of excluding students (rather than staff for whom I would support mandatory vaccination) who are not or can not be vaccinated for a range of reasons such as ease of access (… international students), cultural or health reasons. In short, a complex approach needs to be taken in this regard”.*(Samantha, academic staff, vaccinated)
*“Whilst on a bureaucratic level, this policy would be aimed at protecting the campus and the workplace… [concerns include a] too close dependence between freedom of managing one’s body, and an individual’s access to basic services”.*(Robert, postgraduate, vaccinated)
*“I’d like to know the mechanism by which the university will enforce a mandate like this. I think it would be proven not workable and would introduce a lot of work at the operational end”.*(Sally, professional staff, unsure about vaccination)

The 438 respondents who opposed UWA mandates were also asked for reasons. The main reasons offered included freedom of choice, issues with vaccines, ethics of mandates, and concerns about provoking anti-vaccine sentiment.

Participants who were *somewhat* opposed often used measured language:
*“While I strongly support and encourage the vaccine and have had it without any doubts myself, I am not keen on mandating vaccinations in places such as educational institutions (which are not also healthcare locations)”.*(Rose, professional staff, vaccinated)
*“Restricts the movement of people and therefore individual freedom. University is a public institution not a private company or residence”.*(Bob, academic staff, vaccinated but not sure to have booster)

Participants who were *strongly* opposed made more vehement claims:
*“Denying people their education on the basis of a personal choice they made which does not have harmful effects on the community at large … is immoral and reminiscent of segregatory [sic] and apartheid policies, a path if chosen is going to be a black stain on the University’s record for years to come”.*(Jake, undergraduate, not intending to be vaccinated)
*“Education shouldn’t be hampered by medical status”.*(Frankie, undergraduate, undecided about vaccinations)
*“Mandating the vaccine is unconstitutional. Why should we be forced to be vaccinated if it opposes our beliefs. If I can [work from home] then let this be an option and I’ll happily not attend campus. Why should I lose my job and not be able to feed my family because I chose not to take this vaccine!”*(Tiana, professional staff, not intending to be vaccinated)

Strong opponents of a UWA mandate also invoked mandates coercing or discriminating against people; being ethically wrong; being employed as a method of control, and unreasonably forcing people to disclose medical information.
*“What justification can you even make to discriminate against a group of students? It is morally abhorrent to expel students based on personal medical decisions”.*(James, undergraduate, not intending to be vaccinated)
*“This is discriminatory and not backed by evidence. For an educational institution, implementing this mandate would be embarrassing and on all levels wrong. What evidence is their [sic] that even suggests getting the vaccine for most people? So what is the evidence to mandate the vaccine for most people? Forcing this on people is horrendous”.*(Joel, undergraduate, not intending to be vaccinated)
*“A mandate is not a characteristic of a democratic nation such as Australia where the individual choice is respected and appreciated. I believe such mandate would lead to discrimination and inequality”.*(Sasha, postgraduate, unsure about vaccinations)

Some strong opponents also took a position against the vaccination itself.
*“Lots of studies coming out about the vaccine shows more harmful side-effects, but it is quickly hidden in the media and ignored by government officials. I do not trust big pharma, I do not trust the government”.*(Robert, undergraduate, not intending to vaccinate)
*“The CDC and FDA in the US have already acknowledged that this is not a vaccine and will not prevent the “vaccinated” person from contracting the virus and not passing it on. In Australia, the “vaccine” is still a clinical trial with trials sent to conclude in 2023. Clinical trials are voluntary. No-one has the power to order someone else to take part in a clinical trial without consent”.*(Sandra, professional staff, not intending to vaccinate)

Some vaccinated participants strongly opposed UWA introducing a vaccination mandate on the grounds of freedom of choice.
*“[T]he University has (even) less standing than a democratically elected government to make such invasive policies”.*(Tony, academic staff, vaccinated)
*“Do not believe university should be mandating individual liberty decisions. Thin end of the wedge toward a autocratic governance model”.*(Alex, academic staff, vaccinated but unsure about booster)
*“Because to study is to be your choice. You should not be forced to be vaccinated if you wish to study. We are a University—not a State”.*(Margaret, postgraduate, vaccinated but unsure about booster)
*“I don’t believe the University should take on mandates around health/freedom of choice over health decisions independent of a government mandate. I don’t think the University should take on the privacy and administrative burden of enforcing such a decision. I think people should have freedom over the choices they make about their own health and body, and it is not the University’s role to interfere with that”.*(Joan, professional staff, planning a second vaccine dose)

Other ‘general opposition’ included UWA not being a health campus, claims that mandates were inappropriate or unlawful, the claim that they are impractical to enforce, and the claim that mandates cause psychological issues.
*“The campus is such a large area it would likely be impractical to enforce vaccination for attendance on campus”*(Nicki, professional staff, vaccinated, somewhat opposed to UWA mandate)
*“There is no logical reason for it. If you claim to do it in the interest of public health you would also need to ban the sale and consumption of alcohol on your campuses, deny anyone who is a smoker entry to campus. To pretend that you are protecting the health of students, whilst profiting off activities the have proven health risks is highly hypocritical”.*(Matt, postgraduate, planning a second dose, strongly opposed to UWA mandate)
*“We are not a health campus therefore vaccination status should be voluntary as it is for flu, MMR, etc.”*(Rebecca, academic staff, vaccinated, somewhat opposed to UWA mandate)

#### 3.2.3. University Vaccine Mandate Stance as a Function of Demographic Factors

Participants’ university vaccine mandate stance as a function of their demographic variables are reported in Appendix A.

**Gender.** University vaccine mandate stance differed between females and males, chi-square (2, *N* = 2804) = 22.6, *p* < 0.001. More females supported the university mandate (82.8%) than males (76%), and fewer females opposed it (12%) compared to males (18.2%).

**Age.** University vaccine mandate stance differed across the main age categories (18–24 vs. 25–34 vs. 35–44 vs. 45–54 vs. 55–64), chi-square (8, *N* = 2751) = 20.9, *p* = 0.007. Support for the university mandate was higher amongst older respondents, with 86.9% of 55–64-year-olds and 84.4% of 45–54-year-olds supporting the mandate, versus 78.7% of 35–44-year-olds, 77.1% of 24–34-year-olds, and 77% of 18–24-year-olds supporting the mandate. In contrast, opposition was higher amongst younger respondents, with 17.2% of 18–24-year-olds, 16.5% of 25–34-year-olds, and 16% of 35–44-year-olds opposing the mandate, versus 11% of 45–54-year-olds and 9.2% of 55–64-year-olds opposing it.

**Occupation.** University vaccine mandate stance also differed between staff and students, chi-square (6, *N* = 2794) = 31.5, *p* < 0.001. Support was higher amongst staff, with 87% of academic staff and 83.5% of professional staff supporting the mandate, versus 76.7% of undergraduate students and 77.9% of postgraduate students supporting it. In contrast, opposition was higher amongst students, with 18.1% of undergraduate students and 15.9% of postgraduate students opposing the mandate, versus 8.9% of academic staff and 10.6% of professional staff opposing it.

**Health status.** University vaccine mandate stance did not differ between those who reported an underlying health condition versus those who did not, chi-square (2, *N* = 2794) = 0.640, mirroring the finding that people with comorbidities were not more likely to be vaccinated against COVID-19.

### 3.3. Drilling into Attitudes about Vaccines and Mandates

Previous research and analysis has reported confusion and a lack of clarity around mandates, exemptions, and consequences [2,5]. Accordingly, we asked all respondents questions to elicit the factors underpinning their position on mandates, inviting them to re-articulate their support or opposition when we presented mandate consequences in particular ways.

#### 3.3.1. Moral Duty to Protect Others

We sought to ascertain whether participants regarded each individual to have a moral duty to vaccinate to protect those they worked or studied with. The vast majority (86.30%) agreed with this proposition; 9.38% disagreed, and 4.38% neither agreed nor disagreed.

**University vaccine mandate stance.** Whether respondents agreed with the above question differed depending on whether respondents supported a university mandate or not, chi-square (4, *N* = 2878) = 1725, *p* < 0.001. Of the 2285 respondents who supported a university vaccine mandate, 2243 (98.16%) agreed that people have a moral duty to protect others by getting vaccinated, whereas only 11 (0.48%) disagreed, and 31 (1.36%) neither agreed nor disagreed. In contrast, of the 438 respondents who opposed a university vaccine mandate, only 126 (28.77%) agreed that people have a moral duty to protect others by getting vaccinated, 251 (57.41%) disagreed, and 61 (13.93%) neither agreed nor disagreed. Of the 155 respondents who were neutral (neither support nor oppose) with respect to a university mandate, 113 (72.90%) agreed that people have a moral duty to protect others by getting vaccinated, 8 (5.16%) disagreed, and 34 (21.94%) neither agreed nor disagreed.

#### 3.3.2. Feeling Safe on Campus

We sought to understand whether participants might support mandates to preserve their own personal safety, asking if they feel safer coming to campus if others are vaccinated. Of the 2878 respondents, the vast majority (2415, 83.9%) agreed, while 263 (9.1%) disagreed, and 200 (7%) neither agreed nor disagreed.

**University vaccine mandate stance.** Whether respondents agreed with the above question differed depending on whether they supported a university mandate or not, chi-square (4, *N* = 2878) = 1637, *p* < 0.001. Of the 2285 respondents who supported a university mandate, 2204 (96.5%) agreed they would feel safer on campus if others were vaccinated, whereas only 19 (0.8%) disagreed, and 62 (2.7%) neither agreed nor disagreed. In contrast, of the 438 respondents who opposed a university vaccine mandate, only 109 (24.9%) agreed, whereas 234 (53.4%) disagreed, and 95 (21.7%) neither agreed nor disagreed. Finally, of the 155 respondents who neither supported nor opposed a university mandate, 102 (65.8%) agreed, 10 (6.5%) disagreed, and 43 (27.7%) neither agreed nor disagreed.

**Health status.** Responses were analysed in terms of whether or not respondents reported an underlying health condition making them particularly vulnerable to the effects of COVID-19. However, there was no statistical difference between the groups (chi-square (2) = 0.638, *p* = 0.727); 221 (85.3%) of participants with a health condition said they would feel safer coming onto campus, compared to 2194 (83.8%) of participants without; 15 (5.79%) respondents with a health condition, and 185 (7.06%) without neither agreed nor disagreed with the question. Finally, 23 (8.9%) of people with a health condition disagreed that they felt safer coming to campus if others were vaccinated; this view was shared by 240 (9.2%) of participants without. Thus, whether respondents had a health condition did not impact whether they felt safer coming onto campus when others were vaccinated.

We undertook careful analysis of the survey responses of people who reported comorbidity health conditions on the hypothesis that this cohort would be most concerned about catching COVID-19 on campus and therefore support both vaccines and mandates. However, participants who told us they took campus safety seriously were just as likely to be healthy individuals who supported mandates because they worried about passing the disease on to vulnerable loved ones, as many elaborated in the further comments.
*“As someone who has a parent with cancer, it is so important that I am as safe as possible to reduce the risk of me possibly catching the virus as if it is passed on to my parent, they will most likely die”.*(Ashley, undergraduate, vaccinated)
*“3 of my immediate family members are immunocompromised. Mandating vaccinations at UWA will make me feel safer at university and I would not be potentially putting my family at risk”.*(Claire, undergraduate, vaccinated)

Other supporters did, however, invoke their own personal safety.
*“I am feeling worried about attending campus next semester for face-to-face classes and wondering if I should enrol in units I can do completely online”.*(Dakota, undergraduate, vaccinated)
*“Once COVID-19 comes to WA, I would be reluctant to continue teaching classes on campus if students and staff were not required to be vaccinated. This is particularly the case for tutorials and small group classes where social distancing is not always possible due to the size of many of the teaching rooms”.*(Adam, academic, vaccinated)

#### 3.3.3. Exclusion from Campus

In light of research finding public confusion about mandates, we considered it important to understand how participants made sense of the mandates that governments or universities might introduce. We asked all participants to tell us whether they agreed that people should be able to attend campus even if they choose not to vaccinate, inviting respondents to consider whether they would support the *consequences* of a mandate excluding non-compliers from campus.

In total, 765 (26.58%) respondents agreed that people should be able to attend campus even if they choose not to vaccinate, in contrast to 1667 (57.9%) who disagreed, and 446 (15.5%) who neither agreed nor disagreed. The fact that we invoked the consequence of non-compliance with the mandate evidently caused some of the almost 80% of total respondents who said they supported a mandate to walk back from an exclusionary model.

**University vaccine mandate stance.** Whether respondents agreed with the above question differed depending on whether they supported a university mandate or not, chi-square (4, *N* = 2878) = 1317, *p* < 0.001. Of the 2285 respondents who supported a university mandate, 298 (13%) agreed that people should be able to attend campus even if they choose not to vaccinate, whereas 1630 (71.5%) disagreed, and 357 (15.6%) neither agreed nor disagreed. In contrast, of the 438 respondents who opposed a university mandate, 397 (90.6%) agreed, 21 (4.8%) disagreed, and 20 (4.6%) neither agreed nor disagreed. Finally, of the 155 respondents who were neutral (neither support nor oppose) towards a university mandate, 70 (45.2%) agreed, 16 (10.3%) disagreed, and 69 (44.5%) neither agreed nor disagreed.

**Health condition.** Responses to the above question were analysed separately for respondents who reported an underlying health condition. Whether respondents agreed with the above question differed depending on whether they reported having a comorbid health condition or not, chi-square (2, *N* = 2878) = 8.17, *p* = 0.017. Of the 259 respondents who reported a comorbid health condition, only 59 (22.8%) agreed that people should be able to attend campus if they choose not to vaccinate, whereas 171 (66.02%) disagreed, and 29 (11.2%) neither agreed nor disagreed. In contrast, of the 2619 respondents who did not report a comorbid health condition, 706 (27%) agreed, 1496 (57.1%) disagreed, and 417 (15.9%) neither agreed nor disagreed. This was one of the few scenarios in which people with comorbidities *did* indicate a stronger mandate preference compared to respondents without comorbidities: they preferred for people to not be on campus if they were unvaccinated.

#### 3.3.4. Termination of Employment and Expulsion from Study

We reframed the mandate by introducing a more severe consequence. We asked respondents whether “UWA staff and students who refuse to vaccinate should lose their positions”. In total, 1118 (38.9%) of respondents agreed, with a similar proportion (1180, 41%) disagreeing, and 580 (20.1%) neither agreeing nor disagreeing. As with the question about exclusion from campus, the answers to this question demonstrate that strong support for mandates does not translate into equally strong support for their consequences.

**University vaccine mandate stance.** Whether respondents agreed with the above question differed depending on whether respondents supported a university mandate or not, chi-square (4, *N* = 2878) = 775, *p* < 0.001. Of the 2285 respondents who supported a university mandate, 1101 (48.2%) agreed that staff and students should lose their positions if they refused to vaccinate, 654 (28.6%) disagreed, and 530 (23.2%) neither agreed nor disagreed. In contrast, of the 438 respondents who opposed a university mandate, only 8 (1.8%) agreed, 422 (96.3%) disagreed, and 8 (1.8%) neither agreed nor disagreed. Finally, of those who were neutral (neither supported or opposed) with respect to a university mandate, only 9 (5.8%) agreed, 104 (67.1%) disagreed, and 42 (27.1%) neither agreed nor disagreed.

#### 3.3.5. Making Sense of Non-Medical Exemptions

Non-medical exemptions can enable non-compliers to go about their business under a mandatory vaccination policy. All participants were asked whether people should be exempt from vaccine mandates for religious or personal belief reasons. Two thirds disagreed. Only 19.58% agreed, and 14.36% neither agreed nor disagreed.

**University vaccine mandate stance.** Whether respondents agreed with the above question differed depending on whether respondents supported a university mandate or not, chi-square (4, *N* = 2878) = 1054, *p* < 0.001. Of the 2285 respondents who supported a university vaccine mandate, only 213 (9.3%) supported exemptions for religious or personal belief reasons, whereas 1790 (78.3%) disagreed, and 282 (12.3%) neither agreed nor disagreed. In contrast, of the 438 respondents who opposed a university vaccine mandate, 308 (70.3%) agreed, 65 (14.8%) disagreed, and 65 (14.8%) neither agreed nor disagreed. Finally, of the 155 respondents who were neutral (neither support nor oppose) with respect to a university mandate, 44 (28.4%) agreed, 45 (29.0%) disagreed, and 66 (42.6%) neither agreed nor disagreed.

In the further comments, participants were keen to highlight support for medical exemptions. We presumed that participants would understand that individuals with a registered medical exemption would not face the negative consequences of a mandate. However, responses indicated that this knowledge may not be widespread.
*“Those who cannot be vaccinated due to an existing medical condition should not be penalised in any way for not being vaccinated. Other than in those circumstances, vaccines for staff and students should be mandatory”.*(Craig, undergraduate, vaccinated, supportive of mandates)

An overwhelming number of free text responses supported the option of working or studying online if a mandate was to be introduced. This ‘opt out’ does not require a formal exemption, and aligns with the idea that people excluded from campus should not necessarily be excluded from their jobs or degrees.
*“While I do not agree that people should lose their positions if they refuse to become vaccinated, I do think they should not have access to the campus and be required to work from home”.*(Steve, academic staff, vaccinated, supportive of mandates)
*“Staff and students shouldn’t loose [sic] their places if they refuse to vaccinate, but be allowed to teach/work/study from home”.*(Tanya, academic staff, vaccinated, supportive of mandates)

#### 3.3.6. Comparing UWA vs. WA Government Vaccine Mandate Support

In addition to asking respondents about their attitudes towards UWA mandates, we asked about their support for a WA Government mandate requiring everyone attending campus to be vaccinated if they were able to be. Of the 2878 respondents, 1824 (63.3%) strongly supported and 493 (17.1%) somewhat supported a WA Government mandate. In contrast, 310 (10.8%) of respondents strongly opposed, and 120 (4.2%) of respondents somewhat opposed a WA Government mandate. A further 131 (4.6%) neither supported nor opposed a WA Government mandate.

We simplified this information by combining responses into three categories—Support, Oppose, Neutral. Just over 2317 (80.5%) of respondents supported a WA Government mandate, as compared to 430 (14.9%) who opposed it. This closely aligns with the almost 80% support for a university vaccine mandate. However, to assess the extent to which the same individuals supported or opposed university as well as government vaccine mandates, we examined the discrepancy between respondents who supported or opposed a university mandate versus a government mandate.

We found significant discrepancies between people’s support for a university vaccine mandate versus a government vaccine mandate, chi-square (4, *N* = 2878) = 3268, *p* < 0.001. As can be seen in Figure 1, of the 2285 respondents who supported a university mandate, 2246 (98.3%) also supported a government mandate (left bar). Likewise, of the 438 respondents who opposed a university mandate, 394 (90%) also opposed a government mandate (right bar). This indicates that the majority of those who support or oppose a university mandate feel the same about a government mandate. However, of the 155 respondents who were neither supported nor opposed a university vaccine mandate, 51 (32.9%) supported a government vaccine mandate, 22 (14.2%) opposed a government vaccine mandate, and 82% (52.9%) neither supported nor opposed a government vaccine mandate.

As we specifically asked participants who were neutral or opposed to either kind of mandate for their reasons, our qualitative data explains why some people prefer the idea of a UWA mandate, while others prefer a Government mandate. Some believed it was more appropriate for UWA to introduce the mandate because this limited government over-reach:
*“I don’t like the idea of the government implementing such a restriction, but would not be opposed if the university were to do so”*(Sam, undergraduate, vaccinated)
*“I’m fine with non-government entities enforcing rules as they see fit, but the restrictions any government can enforce into the wider population should only be carefully expanded”.*(Jerry, postgraduate, vaccinated)

However, others believed that government should be responsible for introducing mandates, because governments have the legitimacy to make these decisions.
*“It is one thing for the State Government to mandate vaccination as they are lawfully required to govern for the welfare of the state in particular, to make decisions in relation to public health. This is NOT the domain of a university”.*(Stella, ‘other’, vaccinated)

Accordingly, if the government did not decide to mandate, then the university should not follow this path by itself.
*“If the government does not find it necessary to implement a vaccine mandate state-wide, then the university should not implement one either; but my position for this would also depend on how ‘accurately’ the state government has decided against a vaccine mandate”.*(Jonathan, undergraduate, vaccinated)

There were related conflicting views on the university introducing a mandate. Some participants argued that UWA introducing a mandate would take away ‘academic freedom’ from staff and students—or be seen to be doing this.
*“Mandating vaccinations would go against all except for [sic] UWA core values: ‘academic freedom to encourage staff and students to engage in the open exchange of ideas and thought’ and ‘fostering the values of openness, honesty, tolerance, trust and responsibility in social, moral and academic matters”.*(Riley, professional staff, undisclosed vaccination status, opposed to mandates from government and UWA)
*“Worry that adopting a mandate at the university would attract unnecessary anti-academia attention. I can imagine bad faith critics of higher education (i.e., “anti-vaxxers”, right wing agitators) weaponizing such a mandate as proof that universities are anti-working class or somehow part of a larger corrupt system”.*(Toby, undergraduate, vaccinated, neutral regarding government mandate, strongly opposed to UWA mandate)

Others countered that not mandating vaccinations would damage the University’s reputation as a science-based institution and be in conflict with the University’s values.
*“As a science-based institution, we have to lead by example and do what is right for public health. Introduce mandatory vaccinations, of course allowing exemptions on fact-based medical reasons”.*(Pat, academic, vaccinated, supportive of both mandates)
*“Universities, at a time when knowledge is being attacked by beliefs, should actively support social order based on scientific knowledge or they have lost their way. I feel the foundations of several centuries of progress are under attack through social misinformation that is dispersed to undermine evidence based knowledge”.*(Shane, academic, vaccinated, supportive of both mandates)

#### 3.3.7. Support for Other Types of Government Mandate

Whilst exploring attitudes towards government mandates for universities, we wanted to understand how people viewed other types of mandates that the government might introduce. The vast majority of respondents, 2310 (80.3%), agreed that the government should mandate COVID-19 vaccinations for the wider community to work, travel, or attend events/hospitality venues. In contrast, 411 (14.3%) disagreed, and 157 (5.5%) neither agreed nor disagreed.

Responses to the above question were analysed separately for respondents based on their stance towards a government vaccine mandate. Significant discrepancies between respondents’ stance towards a government vaccine mandate for the university versus additional government mandates were found, chi-square (4, *N* = 2878) = 2569, *p* < 0.001. Of the 2317 respondents who supported a government vaccine mandate for the university, 2221 (95.9%) also supported additional government mandates. Likewise, of the 430 respondents who opposed a government vaccine mandate for the university, 366 (85.1%) also opposed additional government mandates. Interestingly, for the 131 respondents who were neutral regarding a government vaccine mandate for universities, 56 (42.7%) agreed that the government should mandate vaccination for other venues, 16 (12.2%) disagreed, and 59 (45%) neither agreed nor disagreed.

#### 3.3.8. Changing the Behaviour of Hold-Outs

The 216 respondents who were not double vaccinated nor willing to be (henceforth ‘refusers’) were asked how two initiatives and two events would impact the likelihood that they would get vaccinated; 214 refusers responded to the questions below.

Refusers were asked to consider how **“A mandate policy where job loss or exclusion from study is a consequence of refusing COVID-19 vaccination”** would impact their likelihood of being vaccinated; 91 (42.52%) said this initiative would not impact their vaccination likelihood, 42 (19.63%) said it would increase it, and 81 (37.85%) said it would decrease it.

The same respondents were asked to consider how **“An information session about COVID-19 vaccines with scientific experts”** would impact their likelihood of being vaccinated; 147 (68.7%) said this initiative would not impact their vaccination likelihood, 32 (15%) said it would increase it, and 35 (16.3%) said it would decrease it.

Refusers were asked to consider **how the state’s border opening** would impact their likelihood of being vaccinated; 171 (79.9%) said this initiative would not impact their vaccination likelihood, 22 (10.3%) said it would increase it, and 21 (9.8%) said it would decrease it. Refusers’ attitudes towards the **international border opening** were similar; 163 (76.2%) said it would not impact their vaccination likelihood, 28 (13.1%) said it would increase it, and 23 (10.7%) said it would decrease it.

#### 3.3.9. Participants’ Own Additional Suggestions to Change Holdouts’ Behaviour

Using the free-text function, both pro- and anti-mandate participants told us about other measures they thought the University should employ to encourage vaccination and to limit the spread of COVID-19. Some thought these measures should replace mandates; others said they should be employed in conjunction.

There was support for advertising and education:
*“Find creative (though perhaps starkly confronting) ways to increase vaccination rates voluntarily”.*(Jenny, academic staff, vaccinated, neutral to government mandates, supportive of UWA mandate)
*“If UWA mandates… make sure there is enough clear information accessible and make sure to engage with all stakeholders as early as possible and give people enough time to comply. Use a carrot, not a stick”.*(Tara, professional staff, vaccinated, supportive of mandates)
*“Before mandating vaccinations, staff and/or students who are choosing not to get vaccinated should be offered an opportunity to attend a vaccination education course. After attending the course, if they still choose not to get vaccinated (within a reasonable time frame)—then they should not be permitted to come on to campus. Many unvaccinated people have not had an opportunity to learn about vaccinations from the right sources—as a University we have a responsibility to break that cycle by offering a means to break the cycle of misinformation”.*(Fay, professional staff, vaccinated, supportive of mandates)

Other participants wanted further health measures:
*“[T]he university should not only be relying on vaccines… This means an immediate survey of all classrooms and offices and libraries for air safety should be conducted. Ventilation and social distancing are important features in the global COVID response”.*(Emma, postgraduate, vaccinated, supportive of mandates)

Some respondents raised issues regarding morale and lack of trust towards management, noting that these could affect the support for, or the successful rollout of, a UWA mandate. Several respondents wrote supportively about this research taking place, and said that they appreciated the opportunity to be consulted and involved. Involving affected groups in vaccinate mandate discussions is one of the key tenets of designing appropriate policy [22].

## 4. Discussion and Conclusions

This mixed methods study of staff and students at Western Australia’s oldest University, conducted well into the pandemic but whilst the State was still COVID-free, adds to knowledge of attitudes towards vaccine mandates in several key ways. It is striking that there are very high levels of vaccination or vaccine intent, aligning with very high levels of support for a University mandate; for a State Government-supported University mandate; and for additional State Government mandates, compared to previous studies of mandate attitudes, for example, with relatively even support and opposition earlier in the pandemic amongst U.S. [23] and German [24] citizens. The high support at UWA is underpinned by high levels of belief that staff and students owe each other a moral duty to be vaccinated, and that vaccine mandates make individuals feel safer on campus. As expected, and echoing studies overseas, people who are not vaccinated and do not intend to be generally opposed these measures.

There are some relevant demographic differences in our findings. Staff were more likely to be double vaccinated or willing to be compared to students, which was likely related to age and reflected public discourse around COVID-19 posing a higher risk for older people. Staff—an older cohort—also supported mandates more than students did, and across the entire study population, older people were highly supportive of mandates whereas a sixth of those aged under 44 opposed them. There was also a gender divide—women were more likely to support mandates and men were more likely to oppose them.

It is important to note that 8.68% of our sample who had received two doses were reticent about a third dose, suggesting that policymakers need to articulate the value proposition of continued doses carefully and consistently.

People with comorbidities are a key focus of vaccination programs. Health officials seek to protect them from infection, given their propensity for serious health consequences. However, our data shows that this cohort is not more likely to vaccinate than people without comorbidities. This may be because people with comorbidities worry that vaccines themselves will exacerbate their health conditions [25]. This group was not more likely to support mandates and did not report stronger beliefs that mandates would make them feel safer coming onto campus. They did, however, show a greater support for mandates that would exclude the unvaccinated from campus.

Different actors can mandate vaccines in different ways, and we explored how participants viewed university mandates, government mandates for universities, and government mandates for other kinds of activities. While all were strongly supported, they were not all supported (or opposed) by the same individuals, with qualitative responses eliciting competing views about the legitimacy of different actors mandating vaccination. Previous local research has found strong support for business or private sector mandates [2]; some of our participants expressed strong views that universities are well-placed to demand evidence-based behaviours from staff and students.

Perhaps the most important finding of this study is that although participants expressed overwhelming support for university vaccine mandates and less than 20% believed religious or personal belief exemptions should apply, this support diminished significantly when it came to punitive consequences. 41% of respondents did not agree that non-compliant staff or students should lose their positions, and only 35% supported this outcome. Even the prospect of excluding unvaccinated people from campus garnered lower support; 42% of participants did not explicitly support excluding unvaccinated people from campus.

Our findings regarding the predicted impact of specific initiatives on vaccine refusers have policy relevance. Excluding the unvaccinated from campus would likely backfire for a large proportion of holdouts, decreasing vaccination likelihood in more people than it would increase it. It should be noted, however, that people’s self-reported intention does not always predict their future behaviour [26]. At the beginning of the pandemic, many Australians reported in surveys that they would not take a COVID-19 vaccine [9,27]. However, with double dose rates >95% nationally at the time of writing, evidently, many of those people went on to change their minds [28]. Nevertheless, the ‘backfire’ effect does need serious consideration. The Chamber of Commerce and Industry reported 39,000 working West Australians being willing to lose their jobs rather than get vaccinated under the State’s employment mandates [29]; media reports indicate many have followed through [30]. Kreps and Kriner [31] found that unvaccinated Americans would opt out of activities where mandates were required rather than getting vaccinated, demonstrating the strength of holdouts’ positions.

Education sessions are often suggested or employed as ways of improving vaccine acceptance [32]; our participants included remarks to this effect. However, the results from vaccine refusers indicate that an educational approach may push equal numbers towards and against vaccination. Again, this should be interpreted within the context of respondents’ subjective expectations concerning the future impact of this initiative on their vaccination behaviour. Governments or organisations seeking to introduce mandates should still offer information sessions, as orienting some non-vaccinators towards vaccination is helpful, and reducing the perception or experience of coercion for *any* individuals affected by mandates is worthwhile.

Further policy lessons include that institutions or governments introducing mandates should emphasise community concerns about catching COVID-19 and becoming sick or transmitting the disease to vulnerable loved ones. Messaging that addresses males directly may also be important, as they were less supportive of mandates in our study; messaging must address and include various cohorts who can see themselves in what is being communicated. It is similarly important to make it clear in all public communications that people with approved medical exemptions are excluded from any potential mandate (and provide clear information on how to access one). Individuals need ample time to comply with a mandate, and institutions must provide easy access to vaccines as part of this. It is worth providing the option for working or studying from home where feasible, even if for a temporary period.

Despite the lessons of our findings for other contexts, our study has a number of limitations that should be considered regarding its wider applicability to other contexts. First, WA faced a unique COVID-19 (non)experience at the time of our study due to non-pharmacological policy interventions; attitudes in other contexts would likely differ. Second, our respondents were also in a setting where the population is familiar with and supportive of mandates used for childhood vaccinations in state and federal contexts [2,8,9]. This may be different in other jurisdictions. Third, the results should be interpreted as findings from an anonymous voluntary survey administered to staff and students from university management, where participation was encouraged but not required. The response rate of our study was approximately 8.50% for students and 25% for staff, and it is possible participants who had strong views about vaccination and mandates (in either direction) may have been more motivated to participate.

Following this research, The University of Western Australia ultimately elected to require all staff, students, and visitors to be fully vaccinated as a temporary (3 month) condition of entry to all University buildings and enclosed structures, announcing the measure on 25 January and implementing it from 7 February. The measure was subsequently withdrawn on 6 May. As of July 2022, Western Australia was deep into its first wave of COVID-19 and 90% of fixed-term/ongoing staff had reported their status to UWA, with 86% reporting being fully vaccinated (including a third dose where eligible); 82% of Semester 1 students have declared their vaccination status, with 99% fully vaccinated (including a third dose where eligible).

## Figures and Tables

**Figure 1 ijerph-19-10130-f001:**
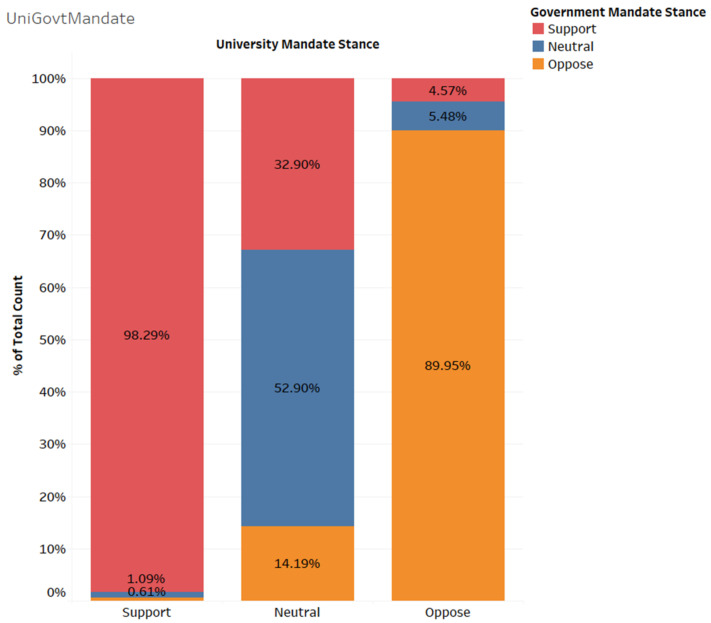
Support for University and Government Mandates.

**Table 1 ijerph-19-10130-t001:** Vaccine Status and Categorisation of participants.

COVID-19 Vaccination Status	Number of Participants	Percentage of Participants	Double Vaccinated or Willing to Be
Double vaxed	2545	88.43%	✓
Will not get vaxed	112	3.89%	
1st dose received, will get 2nd	86	2.99%	✓
Vax status not disclosed	49	1.70%	
Undecided about vax	47	1.63%	
Willing to get 1st dose	31	1.08%	✓
1st dose received, won’t get 2nd	6	0.21%	
Medically exempt	2	0.07%	

**Table 2 ijerph-19-10130-t002:** UWA Mandate Support.

Categories	Number Participants	Percentage of Participants
Support	2285	79.40%
Oppose	438	15.22%
Neutral	155	5.39%

## Data Availability

Quantitative data for this project is available on the Open Science Framework, accessible at: https://osf.io/q96zp/?view_only=3d55e08f505c4d5787b2e1d0e850c9ad.

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
