# Peer review of "COVID-19 Vaccine Mandates: Attitudes and Effects on Holdouts in a Large Australian University Population"

_ijerph, 2022, doi:10.3390/ijerph191610130_

Round 1

Reviewer 1 Report

Comments

This manuscript makes an interesting read. Below are some comments and suggestions to improve the overall quality

Results

·         I suggest the demographic characteristics of respondents be presented in a tabular form as this makes visualization easier and more reader friendly.

·         On lines 158-159: “We classify people willing to have a first dose as willing to have a second…” Why did you classify people willing to have a first dose as willing to have a second since you know there are those who refuse to have a second dose after receiving the first (you have information of such people in your table)?

·         Again, the information on vaccination status by demographic variables will be more reader friendly if presented in a tabular form. The same goes for section 3.2.3. (University Vaccine Mandate Stance as a Function of Demographic Factors)

·         Lines 583-585: “91 (37.8%) said this initiative would not impact their vaccination likelihood, 42 (17.4%) said it would increase it, and 81 (33.6%) said it would decrease it. In this submission, over 11% of responders remain unaccounted for (percentages do not add up to 100%).

·         Kindly account for these responders to make the picture complete.

Reviewer 3 Report

Thank you very much for the invitation to review the paper “COVID-19 Vaccine Mandates: Attitudes and Effects on Hold-outs in a Large Australian University Population”. I believe you have interesting material, but methodologically wrong. Three main questions led me to conclude this:

1. The questionnaire does not have the necessary validity:

The method of collecting data in a survey must be planned so that the procedures determined can guarantee reliability indicators. This decision will depend on the design of the research and the selection of adequate and precise measurement instruments.

The validity of an instrument occurs when its construction and applicability allow a faithful measurement of what is intended to be measured, that is, if the content of an instrument effectively analyzes the requirements to measure the phenomena to be investigated.

In the case of the study, the authors do not do this. No validation is described, only an adaptation to the reality of the authors' country.

The validity and/or calibration of an instrument is conditioned not only to the adaptation of content (as the authors did), but to the number of times it is tried and the analytical procedures, including multivariate statistics. There are several studies that reinforce the need for combination with more consistent and robust analyses, such as criterion and construct, in order to gather more evidence on validity. Reliability is also something that should be added to the results: See: Souza AC, Alexandre NMC, Guirardello EB. Psychometric properties in instruments evaluation of reliability and validity. Epidemiol Serv Saude. 2017 Jul-Sep;26(3):649-659. English, Portuguese. doi: 10.5123/S1679-49742017000300022. PMID: 28977189; https://doi.org/10.1590/0102-311X00143613; Reichenheim ME, Hökerberg YH, Moraes CL. Assessing construct structural validity of epidemiological measurement tools: a seven-step roadmap. Cad Health Publica. 2014 May;30(5):927-39. doi: 10.1590/0102-311x00143613. PMID: 24936810. The adaptation of content is an important step, but it must be added to other procedures and even to a more representative sample for the convergence of objectives, that is, that the instrument measures what it really intends and or was intended to measure.

2. The data collection procedure is unclear and leaves several questions open:

Dissemination procedures; Development and testing and Contact mode;

Tool where questionnaire was made available, ways to prevent double answers; open and closed questions and other security criteria;

Security issues: IP check; Cookies used;

Response rates: Unique site visitor; View rate (Ratio of unique survey visitors/unique site visitors); Participation rate (Ratio of unique visitors who agreed to participate/unique first survey page visitors); Completion rate (Ratio of users who finished the survey/users who agreed to participate)

Among others…

3. Qualitative data are inaccurate:

The method sticks to a formal description without highlighting how the issue is tackled; the relationship between the study site, the sample and the universe, the data saturation paths and criteria, among other details that provide robustness to the study are absent. There is also no deepening of the research instruments, fieldwork and the way in which the analysis was carried out and subsidized.

The presentation of the results cannot depart from the simple description of the empirical data, except for a formal categorization that is not explained. What criteria were used to select the speeches? What is the process of making speeches?

The analysis process and results should be described in sufficient detail so that readers have a clear understanding of how the analysis was carried out, its strengths and limitations.

About the use of NVIVO, What data were used to carry out this analysis? Was the data processed in the Software or was the data apprehended in the forms? Where did the authors extract the key expressions, the central ideas?

Round 2

Reviewer 3 Report

Thank you very much for the extensive review of the material.